# A Review of Performance Prediction Based on Machine Learning in Materials Science

**DOI:** 10.3390/nano12172957

**Published:** 2022-08-26

**Authors:** Ziyang Fu, Weiyi Liu, Chen Huang, Tao Mei

**Affiliations:** 1School of Computer Science and Information Engineering, Hubei University, Wuhan 430062, China; 2Hubei Software Engineering Technology Research Center, Wuhan 430062, China; 3Hubei Engineering Research Center for Smart Government and Artificial Intelligence Application, Wuhan 430062, China; 4School of Materials Science and Engineering, Hubei University, Wuhan 430062, China; 5Hubei Collaborative Innovation Center for Advanced Organic Chemical Materials, Wuhan 430062, China; 6Key Laboratory for the Green Preparation and Application of Functional Materials, Wuhan 430062, China

**Keywords:** machine learning, materials science, performance prediction, deep learning

## Abstract

With increasing demand in many areas, materials are constantly evolving. However, they still have numerous practical constraints. The rational design and discovery of new materials can create a huge technological and social impact. However, such rational design and discovery require a holistic, multi-stage design process, including the design of the material composition, material structure, material properties as well as process design and engineering. Such a complex exploration using traditional scientific methods is not only blind but also a huge waste of time and resources. Machine learning (ML), which is used across data to find correlations in material properties and understand the chemical properties of materials, is being considered a new way to explore the materials field. This paper reviews some of the major recent advances and applications of ML in the field of properties prediction of materials and discusses the key challenges and opportunities in this cross-cutting area.

## 1. Introduction

Materials science, which has experienced a series of stages (from natural materials to independent synthesis, from direct usage to the study of their components, from common technology to artificial intelligence), occupies a crucial place in human history. Today, more and more emerging materials (graphene [1,2,3,4,5], perovskite [6,7,8,9,10], etc.) with special structures (MOF [11,12,13,14,15], COF [16,17,18,19,20], etc.) are appearing on the horizon, and their properties are yet to be explored and improved. However, with the advancement in the research field, materials are repeatedly constrained by many factors such as the complexity of experiments, long inspection times, cost of trials and errors, etc. [21,22,23]. For these reasons, scientists are exploring whether it is possible to find correlations from abundant characteristic data to accurately simulate and predict the properties of substances. Out of consideration, many scientists decided to seek guidance from other fields. Therefore, the combination of computer technology and materials science came into existence.

As applied computational research with the ability to perceive and mimic humans, artificial intelligence (AI) can obtain mapping relationships from various pieces of information and make rational decisions from a huge amount of input and output data [24]. Therefore, machine learning (ML) is well known, and this mapping relationship is identified as feature representation [25]. Today, ML has evolved as the basis for almost all modern applications [26], with three categories: supervised machine learning, unsupervised machine learning, and reinforcement learning. Supervised machine learning refers to training data that already contain explicit categorical information; the mapping relationship between them is feature information to categorical information, and it is widely used for classification and prediction tasks. Unsupervised machine learning refers to data that do not contain explicit categorical information; features are extracted from the information by means of algorithmic models and other means, and the mapping relationship between them relies on algorithms, which are widely used for user clustering and dimensionality reduction. Reinforcement learning refers to methods that maximize the benefit of a model by learning certain states or behaviors in the environment. Each of them has its own merits and demerits, and all have the potential to contribute to the development of materials. For example, supervised machine learning is often used for classification or prediction. Image recognition is a typical example of supervised ML [27,28,29]. On the MNIST dataset, ML attempts to obtain a mapping relationship between pixels and labels. If the two-dimensional image in this example is extended to a three-dimensional space, ML algorithms can also tap into certain features that exist between the material molecules, which can be used in structural design, performance prediction, and other aspects of materials. From a large number of images, it can infer the underlying patterns. Then, an unfamiliar image is identified and labeled according to some feature information. However, it is difficult and time-consuming to extract the right features from raw data that are easily accessible to humans. Therefore, the aim of many scientists has been to find a way to identify them quickly and extract them accurately.

As one particular type of machine learning, deep learning (DL) plays a major role in this field [24]. It can quickly extract effective features from multi-dimensional data and use its complex neural network structure to transform simple features into abstract ones, thereby learning the intrinsic patterns and levels of the sample data. The information obtained from these learning processes can be of great help in the interpretation of data such as text patterns, image patterns, etc. [30].

In recent research, various types of derivative algorithms represented by artificial neural networks (ANNs) have been widely used in various fields in several disciplines. Shallow ANNs are rapidly emerging into various disciplines with their advantages of portability, good understanding, and high performance, such as Tohidi et al. [31] using a three-layer backpropagation neural network (BPNN) to predict the residual buckling strength of damaged members. In addition, deep ANN network structures have been created to be able to extract more features from the data, which have more hidden layers to make more accurate predictions, such as in the work by Bhaduri et al. [32], who built a deep CNN to predict the stress field for systems with a larger number of fibers using CNNs that were trained on data from relatively cheaper systems with a smaller number of fibers. Researchers have continued to build more complex networks in order to improve their performance, such as Fang et al. [33,34], who proposed a deep belief network (DBN) framework for studying the structural performance of cold-formed steel channel sections and prediction of axial capacity. Therefore, it can be seen that deep learning performs well in fast feature extraction and can guide the design of various novel materials.

With the maturation of ML algorithms, significant progress has been made in their integration with the field of materials science, such as in the design of battery materials [35,36,37,38,39,40], nanoporous materials [41,42,43,44,45,46], etc. Through ML methods, the properties and structures of materials can be predicted quickly and accurately, which has inspired innovation in the design and degradation prediction of high-performance materials. By using suitable AI algorithms to rationalize the design and discovery of materials, new materials and structures can be systematically discovered, and the waste of resources and environmental pollution caused by numerous non-essential experiments can be effectively reduced. This can have significant social value [30]. In addition, a reasonable prediction of material degradation not only leads to efficient utilization and real-time renewal, but also promotes the industrialization of new materials. In this paper, a brief overview of the current applications of machine learning in the field of materials performance prediction is presented. The primary work focuses on the prediction of the performance of the data and materials, which is shown in Figure 1. The method used is explained in detail, and a summary and outlook on the current difficulties faced in these crossing fields are elaborated.

## 2. Data

Data are the cornerstone of all performance predictions. The acquisition, generation, and preprocessing of data are crucial for ML models. A huge amount of data is a prerequisite for the effective application of ML models, and high-quality data have a direct impact on the generalization ability of the ML models. Data become a key link between ML models and materials when they are combined. Currently, there are two primary ways for researchers to obtain the data: (1) collecting data from the literature or existing databases; (2) generating their own databases from high-throughput experiments or simulations (as illustrated in Table 1) [24]. If the data obtained are too simple, it is unnecessary or difficult to use ML methods for solving the corresponding problems. However, the researcher can always obtain data that are not sufficient to express the overall features. So, researchers usually use suitable methods to design features appropriately and control the size and distribution of the dataset to be generated, which is a well-known method of data generation. At the same time, not all available data can contribute positively to the prediction, so it becomes essential to preprocess the data to exclude some of the interfering ones.

### 2.1. Data Collection and Generation

In 1999, the concept of materials informatics was pioneered by Professor John R. Rodgers. The aim was to use theoretical simulations to try as many real or unknown materials as possible with known experimental data [47]. Another aim was to build databases of their composition, structure, and various physical properties, along with exploring the relationship patterns between material composition, structure, and properties through data mining to obtain structural predictions of optimum performance. Exploring the use of the concept of “building units” and “high-throughput sifting” in combinatorial chemistry for computer simulations of materials to build new compounds has become a hot topic of interest in the industry, both at home and abroad. Within this background, “high-throughput computational materials”, a simple yet powerful concept, was born. Considering the three aspects (quantum mechanics–thermodynamics calculations, large amounts of data, and intelligent data mining), a large database of thermodynamic and electronic properties about existing and hypothetical materials was created, which could be intelligently queried to search for materials with the desired properties [48]. The appearance of this design has led to the creation of several databases such as AFLLOW, Materials Project (MP), MATDAT, MatWeb, MatMatch, MakeItForm, and MatNavi (Table 1). These databases contain a variety of material information, such as mechanical properties, constituent units, electronic characteristics, etc., all of which provide favorable data to support the prediction of material properties and structures. Apart from obtaining data from the databases, searching the literature for relevant material information can also be treated as a method. Although not as comprehensive as the database system, the data obtained in this way are more relevant and provide direction for synthesis and application.

While conducting computer experiments or simulations of high-throughput materials, researchers have a great deal of freedom to design the size and distribution of the dataset, i.e., the features of the generated dataset, which allow for more possibilities in the generalization performance of the ML models. Current research focuses on balancing datasets by improving the generalization performance of models. For instance, with only 228 experimentally synthesized zeolite frameworks in the IZA database, Deem and coworkers [49] devised an algorithm to generate thousands of hypothetical pure silica zeolites and consequently reported the most feasible zeolite topologies. Yang et al. [50] used the Gaussian random field (GRF) method to create a synthetic microstructure image dataset of materials with different compositions and dispersion patterns. In addition, there are several common means of generating data today, such as the use of the finite element method (FEM) to generate high-contrast composites [51], two-dimensional (2D) mosaic composites [52,53,54], as well as microstructure [55,56,57,58] and datasets [59,60,61,62,63] of three-dimensional (3D) materials. The existing studies suggest that combining diverse knowledge from materials science, solid mechanics, and other related fields can generate datasets that are more representative of the design space and thus give better results with the applied models.

### 2.2. Data Preprocessing

To predict the ML models more accurately, the data should have more distinctive features to obtain better generalization performance. Thus, the dataset must be processed in advance. By reviewing the previous literature [64,65,66,67], it can be observed that the current methods of data preprocessing can be divided into two main categories: (1) manual processing of data based on data transformation and data augmentation; (2) feature extraction of data based on ML preprocessing methods.

Recently, Kojima et al. [68] optimized material images by using image-processing techniques as a preprocessing technique to increase the initial dataset, which was insufficient for the number of ML model analyses (Figure 2a). Using this data enhancement method, 256 images were created from one filled shape, and from 45 filled shapes, 11,520 datasets were prepared, providing a large source of useful data for rubber tires in fracture performance prediction. Moreover, Yildirim and coworkers [69] also analyzed 5508 pieces of experimental data collected from the literature on methane vapor by means of decision trees in ML preprocessing. With 21 variables related to catalyst preparation and operating conditions as input variables, correlations and trends between variables were extracted to provide an idea for finding the optimal conditions of catalytic methane. Thus, data enhancement and ML methods are quite crucial for data preprocessing. Comparatively, data enhancement requires relatively more manual transformations and practical operations; however, ML methods can directly parse the data into abstract features, reducing human involvement. In practice, most researchers use these two methods comprehensively. For example, Dunn et al. [70] performed several operational steps (such as feature extraction, feature reduction, model selection, hyperparameter tuning, etc.) on their proposed Matbench base test suite and used the ML model in the suite to accept the composition and crystal structure parameters of the materials in the dataset and then return the predicted results (Figure 2b).

## 3. Performance Prediction

### 3.1. Material Properties

With the further enhancement in the fields of ML and DL, along with a fast increase in computational power, ML models have entered and been applied in many different fields [25,71,72], including material performance evaluation. Through data feature extraction and the fitting of data correlations, ML models can quickly learn to acquire new skills or new knowledge to efficiently and accurately predict mechanical performance and explore new components or structures better than the training data in the design space [73]. Not only does this reduce the wastage of resources caused by repeated trials, but it also improves the efficiency of screening and searching for new materials and structures. One of the most widely used ML models for performance prediction is the artificial neural network (ANN) [74].

ANN is a parameter generalization function approximation that belongs to supervised learning, i.e., feature labeling learning. Neural networks are usually based on multilayer perceptron (MLP) networks, whose basic structure comprises at least three layers: an input layer, an output layer, and one or more hidden layers [75]. The structure of an MLP is demonstrated in Figure 3. Assuming that the MLP has *n* inputs and an activation function *f*, the transfer output *y* of a neuron is defined as a weighted sum; *w_i_* denotes the connection weight of the neuron; *b* represents the bias value. So, the next neuron can be represented as follows [76]:$$y=f\left(\overset{n}{\underset{i=1}{\sum }}\left(w_{i}x_{i}+b\right)\right)$$

After multiple neurons are passed and accumulated, the initial randomly assigned weight vector is automatically updated by gradient descent when the information is passed from the input layer to the output layer while minimizing the loss function on the training set until the loss stops reducing. In general, with the increase in the number of hidden layers and their units in a neural network, the fitting effect of the neural network is enhanced. Therefore, it can be used to map any type of output and to achieve continuous optimization of the weight parameters and cost functions to approximate the actual data by fitting several parameters to the data. At the same time, an excessive number of network layers and neurons will lead to the risk of overfitting, which emphasizes certain useless weights to skew the predictions from normal, and they can be solved by simplifying the model or ending it early as usual. Currently, there are several variants of ANN, such as convolutional neural networks (CNN), recurrent neural networks (RNN), long short-term memory (LSTM), etc. [77,78,79,80,81]. Each of these networks is applicable for some particular fields and categories.

Similarly to the connection of synapses in the brain, ANN relies on the interconnected relationships of internal nodes and continuous fitting to achieve the purpose of processing information. Since ANN can mine features from a large amount of experimental data and can solve complex nonlinear as well as multi-dimensional functional relationships without prioritization, it is widely known and applied to material property prediction and design. As the following shows, there are several typical applications of ANN to the design of novel materials.

#### 3.1.1. Nanomaterials

Nanomaterials acquire many distinctive effects because of their tiny particle size, such as interfacial effects, small-size effects, quantum-size effects, and macroscopic quantum tunneling effects. As a result, nanomaterials have many properties that are different from or even opposite to homo materials with other sizes and can also be changed with the changes in size. Therefore, the exploration and design of different nanomaterials is always of great concern for many scientists.

Considering the storage properties of nanomaterials, Lee et al. [82] used ANN to generate shapes of crystalline nanoporous materials in energy space and performed extensive molecular simulations to identify the performance limits (Figure 4a). To break through the complexity of the ANN method for building crystalline nanoporous materials, the team innovatively used an abstract representation of such materials based on the potential energy space of the target gas molecule (in the case of ANGV, methane) instead of building a conventional material space on an atomic/molecular basis. In the construction process, Lee et al. proposed a new type of GAN network: energy shape GAN (ESGAN). The structure of the network is demonstrated in Figure 4b. There are two networks in this model with the discriminator D and the generator G. The discriminator D helps to distinguish the real and simulation-generated zeolite 3D methane energy lattices caused by ANN. The generator G, which includes two sub-generators, the energy lattice generator, and the lattice constant generator, is used to generate increasingly realistic methane energy lattices to cheat the discriminator for repetitive learning. Additionally, the team added an extra layer and extra function to the discriminator for predicting the lattice constants from the fractional energy lattice. The loss functions of the discriminator and generator are denoted by $L_{GAN,D}$ and $L_{GAN,G}$, respectively, and the loss function $L_{FE-matching}$ is used for free energy matching, which helps the generator to create suitable energy values. Further, the desired simulated performance prediction is obtained by equation calculation and simulation. Apart from this, the team also explored the validity of the method by selecting silica crystalline zeolites as the materials for evaluating the storage limitations of methane. By comparing the distribution of the generated lattice constants with the real zeolite material, excellent consistency was observed in all performance dimensions (Figure 4c,d), which indicates the prominent accuracy of this prediction method while providing new ideas and methods for the performance prediction of other gas molecules and nanomaterials.

#### 3.1.2. Adsorbing Materials

With the worsening of environmental problems, material recycling has always been an urgent issue that must be addressed. Therefore, along with the prediction of storage performance, the performance of the material during recycling also needs to be researched.

Recently, Franco et al. [83] obtained high purity indium concentrate from liquid crystal display (LCD) recycling by ML simulation prediction and a low-cost simple adsorption experiment to obtain the optimal mixing ratio for maximum indium adsorption. In this work, Franco et al. combined ANN and an adaptive neuro-fuzzy inference system (ANIFS) to perform a composite analysis of indium(III) adsorption and observed that among the 10 different adsorbent materials tested, chitosan had the best adsorbent performance ($1000\;\mathrm{mg}/g^{-1}$ of adsorption was achieved within 20 min) (Figure 5). On the one hand, ANN can learn from experimental data to solve complex nonlinear, multi-dimensional functional relationships without prioritization; on the other hand, ANFIS can mix the learning effect of ANN with the inference effect to obtain more realistic results. By using specific surface area (AS), zero charge point (pHpzc), adsorption time (t), and adsorbent dose (D0) as input variables, the structures can be established. The ANN system, which is constructed using the Levenberg–Marquardt algorithm [84], can be divided into three groups with a ratio of 7:1.5:1.5 for the training set, test set, and validation set, respectively. The structure of the ANN contains four hidden neurons, as shown in Figure 5. The data in the ANFIS system can be divided into two groups with a 7:3 ratio between the training set and the test set, and the test set is checked using the “randperm” function. The ANFIS architecture contains three affiliation functions for AS, two affiliation functions for pHpzc, two affiliation functions for t, and three affiliation functions for D0, and generates a set of rules. Using R2, MSE, SSE, and ARE as the performance evaluation metrics of the model, the chitosan was simulated as the best adsorption material for indium (III). During the actual experiments, it was observed that the low dose of chitosan could maximize the adsorption of In(III) (from 0.6 mg g^−1^ (original LCD) concentrated to 1000 mg g^−1^) in a short time by comparing the IR spectra of different adsorption materials before and after the adsorption of indium(III), which was in accordance with the simulation results. Researchers also investigated the mechanism based on experimental results, leachate morphograms, FT-IR, and the literature, and proposed the result of the possible coordination linkage between In and chitosan. According to the theory, each chitosan glucosamine unit can adsorb one $In^{3+}$ with the help of $SO_{4}^{2-}$ to form a coordination complex. It can explain the high affinity of chitosan with In(III) ions and the high adsorption capacity of the material in a short time, guiding the efficient recycling of different kinds of materials and high-adsorption performance material design.

#### 3.1.3. High-Performance Materials

High-performance materials are present in all aspects of human life due to their lightweight [85,86,87] and high mechanical properties [88,89,90,91,92]. Efforts to explore the improvement of material properties have never stopped and are continuing in an efficient and rational way.

Recently, Mentges et al. [93] built a neural network model for predicting the elastic properties of short fiber reinforced composites (SFRC) [94], and obtained a training set and validation set dataset for ANN by using a micromechanical simulation approach. Before building the ANN model, to ensure that the dataset could enable the ANN model to achieve the best results, Mentges et al. used parameter space evaluation for magnifying the spatial distribution of the dataset to improve the consistency of the dataset and avoid overfitting. Apart from that, it must be checked if the model parameters are uniformly distributed in the dataset to ensure the accuracy of the simulation. The ANN model contained 12 independent input variables of basic fiber properties and 21 independent variables to describe the stiffness tensor of the loaded material. By testing several different neural networks, Mentges et al. determined that the ANN model consisted of three hidden layers with 25 neurons per layer and an exponential activation function, with RMSE used as a performance metric. The Adam optimizer with a learning rate of 0.05 and exponential decay rates of 0.95 and 0.99 for the first and second order moment estimates, respectively, was used. Luckily, it was considered a good fit when the minimum RMSE on the test dataset was only 0.0046. It can be inferred from the experimental results that the mean-field predictions at fiber volume fractions of 0.13 and 0.21 were in good agreement with the experimental results, but there was a large overestimate of the mean-field prediction at the highest fiber volume fraction of 0.29 (Figure 6). The results suggest that the simulation range was somewhat limited in terms of fiber volume fraction, but it provided guiding ideas in terms of research and simulation to predict the mechanical properties of the material.

In addition to the above introduction, ANN is also widely used in the simulation and prediction of other properties of materials such as compost maturity efficiency [95], sizing the catalyst synthesis [96], and predicting electronic structure image arrays [97]. With the rapid development of machine learning, there are many machine learning methods (particle swarm optimization [98], deep deterministic policy gradient [99], etc.) that can be used for outstanding contributions to material property prediction and structure design.

### 3.2. Degradation Detection

With the development and usage of materials, there is a key concern for the industry about service life and degree of degradation. Especially for applications in the battery field, accurate direct measurement of the chemical processes occurring within the materials deployed in the field is largely unfeasible. Therefore, the effective detection of material degradation has also become a pressing issue in ensuring efficient and stable product operation. Machine learning, through the extraction of features and the analysis of correlations between data, can be used to predict the deterioration of materials, not only to ensure the efficient use of materials but also to facilitate the industrialization of new materials.

For example, Wang et al. [100] proposed a stacked long and short-term memory (S-LSTM) model with two dropout parameters to fit the degradation system of a proton exchange membrane fuel cell (PEMFC), which can be constructed by voltage or power versus cycle time. Therefore, the stackable LSTM architecture could improve the prediction accuracy of fuel cell degradation and optimize the hyperparameters of the S-LSTM model by a differential evolutionary algorithm. To evaluate the performance of the proposed model, Wang et al. selected two PEMFC systems, FC1 and FC2, from the FCLAB Research Federation for demonstration and as data sets. They reconstructed the original datasets using the mean of each variable for each hour, generated 1155 and 1021 sample datasets on the two systems, and used MAPE and RMSE as performance measures of the model. Then, they proposed the S-LSTM based model, which contains two layers of LSTM. The memory unit of LSTM generally consists of three nonlinear gating units, that is, the forgetting gate, the input gate, and the output gate, and the structure of its memory unit is shown in Figure 7a. In order to reduce overfitting and improve the model performance and generalization ability, Wang et al. designed two dropout parameters and used the DE algorithm to determine the optimal parameters of the S-LSTM model. The framework of the overall model is shown in Figure 7b. According to the results, the S-LSTM model outperforms the other models in the remaining useful life (RUL) prediction of the degradation of PEMFC in terms of the mean absolute percent error and the root mean square error, indicating that the S-LSTM model can effectively provide high prediction accuracy for the prediction of total voltage degradation. In addition, they also provided an outlook on the development of this model: further prediction of RUL uncertainty by Monte Carlo dropout or Bayesian methods will be investigated in the future, while more deep learning models with different network structures need to be developed to improve the degradation prediction of PEMFC systems, which is shown for the use of deep learning models for online RUL prediction.

Then, Li et al. [101] proposed a deep learning model with multiple LSTM layers in both encoder and decoder blocks to monitor battery health, which can predict the degradation trajectory of the battery at once without iteration or feature extraction, containing the life inflection point and the end point. The capacity time series up to the current point is fed into the constructed model by collecting slight data from laboratory experiments or battery management systems, and then the model calls out the future capacity time series of the battery up to the end of life (EOL). Especially, because the battery degradation history can be considered as a variable-length sequence, the size of this input sequence increases as the battery is used in this model (Figure 8a), and performance improves over time as more data become available. In this task of battery degradation prediction, the authors employ two multilayer LSTM architectures working in concert. The encoder is responsible for encoding the input sequence into a static embedding vector, and the decoder processes the embedding vector and provides the output sequence (Figure 8b). Experimental results show that the model is able to predict the degradation trajectory of the capacity accurately and adapts well to time series. Compared to state-of-the-art approaches, the one-shot approach shows an increase in accuracy and in computing speed by up to 15 times. This work further highlights the effectiveness of DL approaches in the domain of degradation prognostics.

In addition to the battery sector, machine learning plays an important role in the prediction of degradation of other materials. For example, to estimate the photocatalytic decomposition of PFOA on various photocatalysts, different ML algorithms, including multiple linear regression (MLR), random forest (RF), ridge regression (RR), multilayer perceptron (MLP), gradient boosting machine (GBM), adaptive boosting (AdaBoost) and support vector machine (SVM), were used by Li et al. [102] to nominate a potential and effective method. After considering numerous factors, such as solution pH, solution temperature, catalyst dose, light irradiation intensity, irradiation wavelength, irradiation duration, initial PFOA concentration, type of catalyst, and oxidizing agents, the GBM model was found to give better results than other models. This experiment not only showed how to predict the degradation of PFOA but also provided new ideas for the application of machine learning in other fields.

## 4. Outlook and Conclusions

This paper reviewed the significant contribution of ML to material performance prediction in two fields (material property detection and degradation detection). Whether in material properties prediction or degradation detection, ML can produce more satisfying prediction results compared to traditional data analysis [103], significantly improve material design efficiency, reduce waste of materials, and shorten exploration time. However, as machine learning techniques continue to be tested in the materials field, many issues are being highlighted and need to be further optimized. They are primarily divided into the following aspects:

**Method and application innovation.** Since the combination of applications between machine learning and materials has not been developed over a long period, previous studies were used to describe the related network parameters, network structures, training algorithms, data for training and testing neural networks, etc. However, with a background in materials science, the ML methods are so outdated that they cannot be satisfied with many application scenarios. For example, most ANN models are based on feedforward neural networks, while different types of neural networks such as time series networks [104,105,106], recurrent neural networks [107,108,109], and adversarial networks [110,111,112] have appeared in the field of computer science. Later, the use of various new neural networks mixed in materials science will become a future trend. Apart from the introduced performance prediction, material identification, mapping analysis, and data optimization are also current directions in application development. Achieving the goal of studying materials quickly and accurately requires continuous innovation in methods and applications in cross-cutting fields.

**Principle exploration.** The prediction of material properties cannot be achieved without using machine learning methods and material formulations. However, there are many principles that are yet to be fully explored. For example, in the field of machine learning, although the structures can be simulated or the properties can be predicted through neural networks, the accuracy of the results needs to be proved by the final experimental results. However, it is difficult to validate the reasons for the success of these structures or algorithms, which are considered “black boxes”. In the field of materials science, the relationship between different data quantities remains to be extensively explored; moreover, the principles of molecular dynamics [113,114,115,116] and density function theory [117,118,119,120,121] required for the calculations have their shortcomings and unexplained parts. Exploring the underlying principles determines not only the efficiency but also the accuracy of machine learning methods applied in the materials field, which must be further developed.

**Data supporting.** Extracting different features from data (including performance data, image information, bonding information, etc.) and fitting them by computation to obtain correlations between different properties are the principles of machine learning. Therefore, data are considered to be the foundation of the whole machine learning application. Research shows that the performance of neural networks depends on the amount of data, and it can be somewhat extended when there is a large amount of data to build an optimized neural network model that enhances the accuracy of the prediction results [103]. Though the amount of data can be effectively expanded by appropriate methods (e.g., data inversion, data rotation, scale transformation) [122,123,124], small amounts of data and few data types still limit the accuracy and validity of the fitting results.

Although the potential of ML in designing new materials has not yet been fully developed, there are several future opportunities and challenges that are yet to be explored and addressed. ML methods in computer science are constantly evolving, with the principles and examples applied in materials science also following. It is believed that in the future, ML-based research methods will be more deeply integrated with traditional experimental investigations to jointly contribute to the development of the materials field.

## Figures and Tables

**Figure 1 nanomaterials-12-02957-f001:**
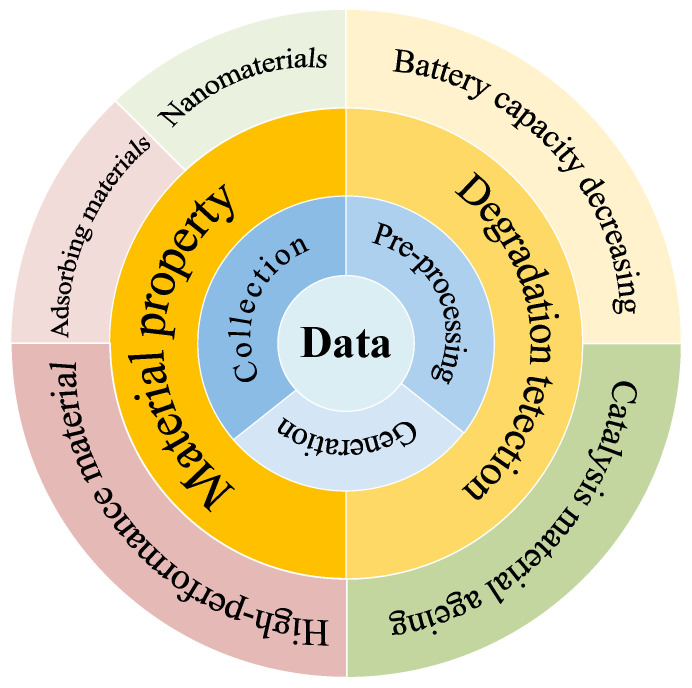
An overview of the components of machine learning for materials’ performance prediction. Data are at the heart of performance prediction, including data acquisition, data generation, and preprocessing. Based on the correlation between the properties derived from the data, predictions can be made for novel material properties (nanomaterials, adsorbing materials, high-performance materials, etc.) and degradation detection (decreasing battery capacity, catalysis material aging, etc.). It is worth pointing out that the predicted fields that can be obtained are not limited to the tasks illustrated above in this schematic, but also include other aspects such as atomic bonding energies, thermodynamic properties, etc.

**Figure 2 nanomaterials-12-02957-f002:**
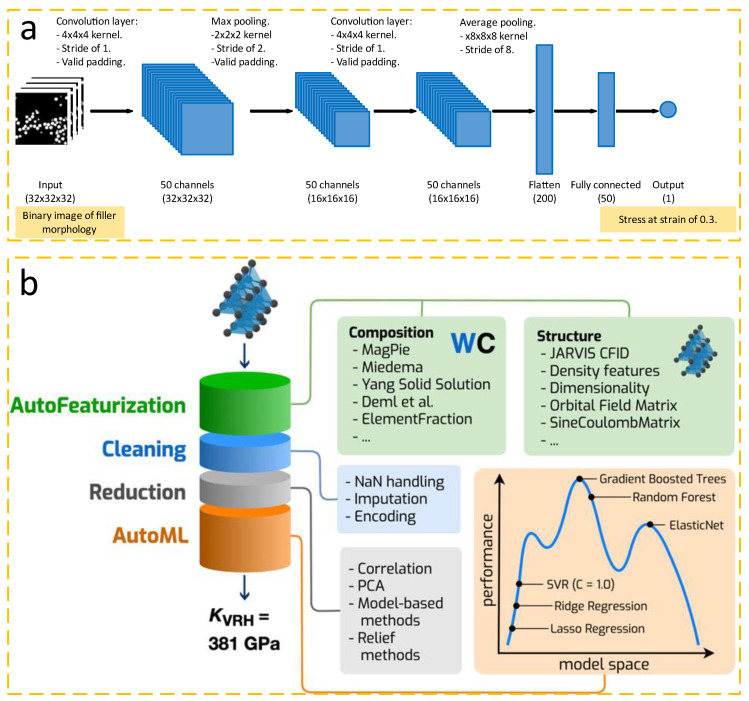
(**a**) CNN architecture. Reprinted from ref. [68]. (**b**) The AutoML+Matminer (Automatminer) pipeline. The pipeline can be applied to composition-only datasets, structure datasets, and datasets containing electronic band structure information. Once fit, the pipeline accepts one or more material primitives and returns a prediction of a material’s property. During auto featurization, the input dataset is populated with potentially relevant features using the Matminer library. Next, data cleaning and feature reduction stages prepare the feature matrices for input to an AutoML search algorithm. During training, the final stage searches ML pipelines for optimal configurations; during prediction, the best ML pipeline (according to internal validation score) is used to make predictions. Reprinted from Ref. [70].

**Figure 3 nanomaterials-12-02957-f003:**
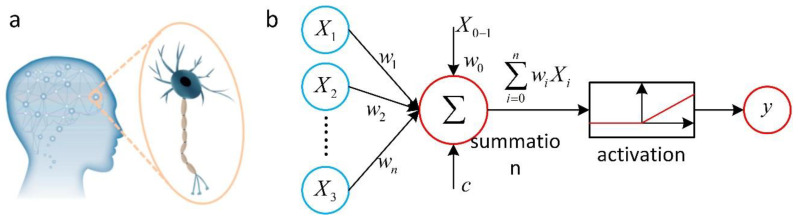
(**a**) Schematic diagram of the biological neural network. (**b**) Flow chart of data processing within a neuron. Reprinted with permission from Ref. [75]. Copyright 2021 Elsevier.

**Figure 4 nanomaterials-12-02957-f004:**
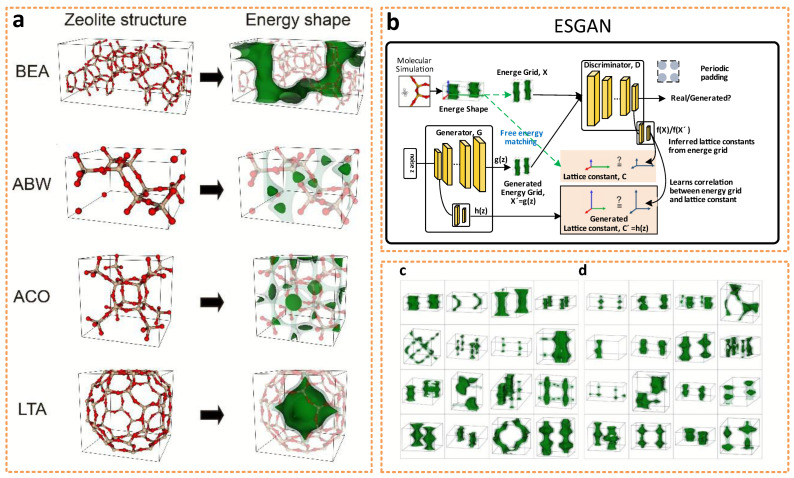
(**a**) Left column illustrates the unit cells of four prominent zeolites (BEA, ABW, ACO, LTA). Right column shows the corresponding methane potential energy profiles (i.e., shapes) computed from molecular simulations (dark green: low energy, light green: higher but accessible energy, white: inaccessible regions). (**b**) Overall schematics of ESGAN. (**c**) Real energy shapes of zeolites from Deem’s database, and (**d**) generated energy shapes from ESGAN. Reprinted with permission from Ref. [82]. Copyright 2019 Royal Society of Chemistry.

**Figure 5 nanomaterials-12-02957-f005:**
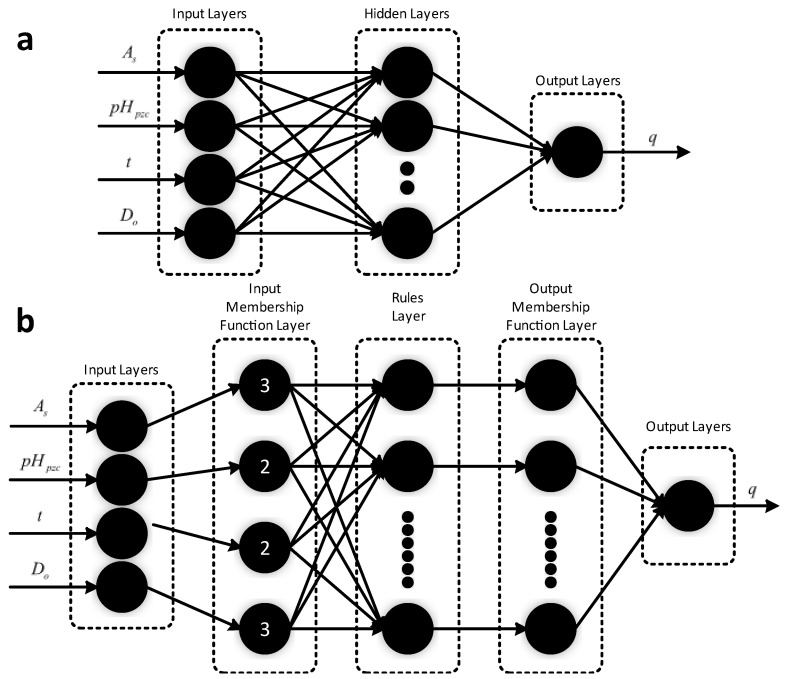
(**a**) Artificial neural network structure. (**b**) A simplified version of the ANFIS architecture. Reprinted with permission from Ref. [83]. Copyright 2020 Elsevier.

**Figure 6 nanomaterials-12-02957-f006:**
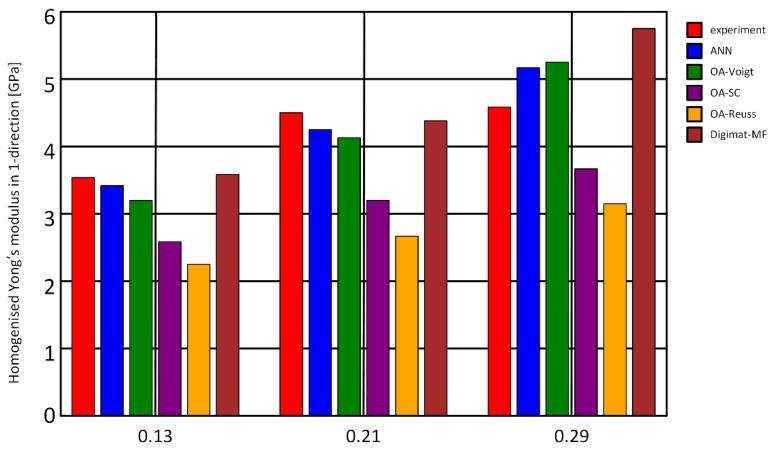
ANN predictions of the flax reinforced polypropylene composite Young’s moduli E11 compared to experiments, orientation averaging (OA) results, and mean-field predictions using Digimat-MF (for three fiber volume fractions 0.13, 0.21 and 0.29). Reprinted from Ref. [93].

**Figure 7 nanomaterials-12-02957-f007:**
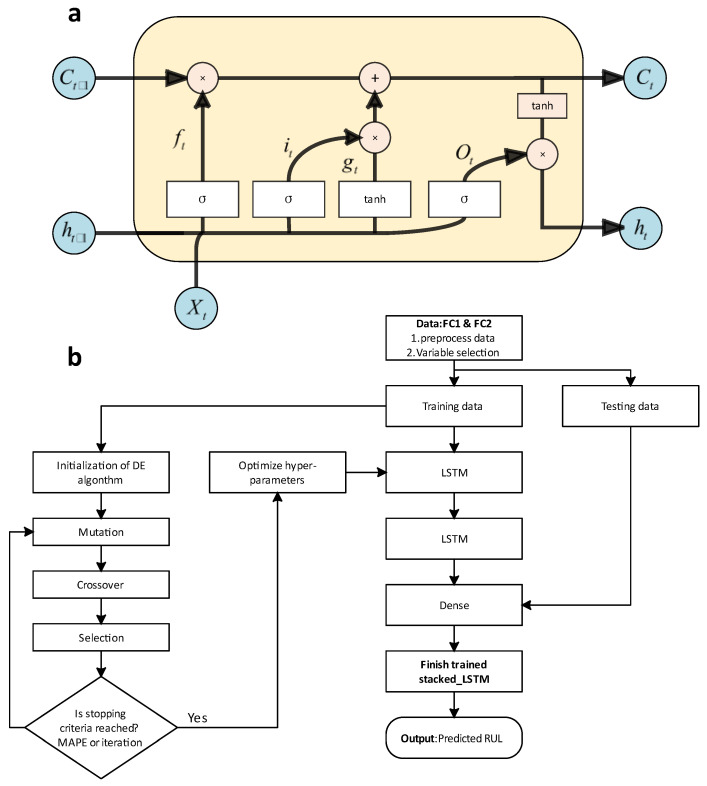
(**a**) The framework of LSTM; (**b**) RUL prediction using a stacked LSTM model. Reprinted with permission from Ref. [100]. Copyright 2020 Elsevier.

**Figure 8 nanomaterials-12-02957-f008:**
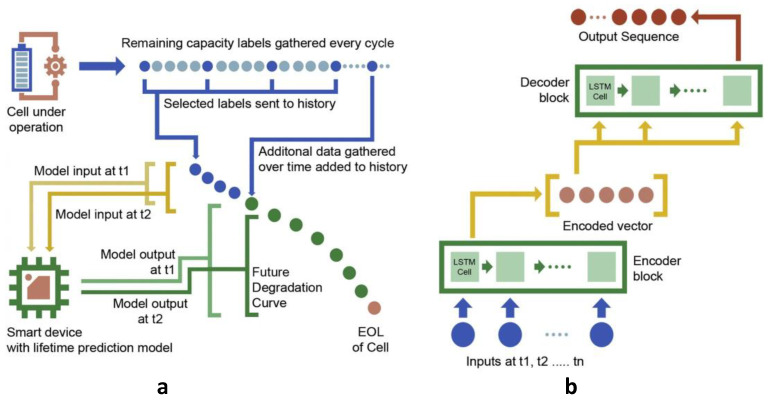
(**a**) Choice of input to the model and the approach of increasing the input window with time. (**b**) The general architecture of the sequence-to-sequence neural network. Reprinted from Ref. [101].

**Table 1 nanomaterials-12-02957-t001:** Popular databases in performance prediction of materials.

Database Name	Material Categories	Features	URL
MatWeb	Metals, plastics, ceramics and composites	Tensile strength, breaking strength, Vicat softening point, etc.	https://www.matweb.com/search/PropertySearch.aspx (accessed on 13 July 2022)
NIST	Metals, polymers, etc.	Thermochemical, thermophysical and ion energetics data	http://webbook.nist.gov/chemistry/name-ser.html (accessed on 13 July 2022)
AZO materials	Alloy, rubber, plastics, etc.	Mechanical strength, element, molecular weight, etc.	https://www.azom.com/materials-engineering-directory.aspx (accessed on 13 July 2022)
M-Base Company	Polymer	Tensile modulus, yield stress and strain, density, molding shrinkage, etc.	https://www.materialdatacenter.com/mb/ (accessed on 13 July 2022)
Ceramic Industry	Ceramic materials	Forming method, sintering process/temperature, tensile strength, bulk resistivity, dielectric strength and elastic modulus, etc.	http://ceramicindustry.com/directories/2718-ceramic-components-directory/ (accessed on 13 July 2022)
NIST ATERIAL MEASUREMENT LABORATORY	Materials	Phase diagram, various thermodynamic and kinetic parameters, atomic spectra, physical parameters, etc.,	https://www.nist.gov/mml (accessed on 13 July 2022)
PoLyInfo	Polymer	Chemical formula, type of material, physical properties	http://polymer.nims.go.jp/ (accessed on 13 July 2022)
CAMPUS	Plastic	Molding shrinkage, breaking strength, Vicat softening point, structure, etc.	http://www.campusplastics.com/ (accessed on 13 July 2022)
Cole-Parmer	Materials	Chemical compatibility	http://www.coleparmer.com/techinfo/chemcomp.asp (accessed on 13 July 2022)
The materials project	Materials	Chemical formula, type of material, physical properties, etc.	https://materialsproject.org/ (accessed on 13 July 2022)
crystalstar	Crystal	Crystal structures of organic, inorganic and metal-organic compounds and minerals	http://www.crystalstar.org/ (accessed on 13 July 2022)
FactSage Database	Materials	Phase diagram and various thermodynamic and kinetic parameters	http://www.crct.polymtl.ca/ (accessed on 13 July 2022)

## Data Availability

Not applicable.

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
