# Peer review of "A Review of Performance Prediction Based on Machine Learning in Materials Science"

_nanomaterials, 2022, doi:10.3390/nano12172957_

Round 1
Reviewer 1 Report
The article refers to the important problem of applying machine learning across data to find correlations in material properties and understand the chemical properties of materials. This approach is recognized as a new way of solving problems related to designing the properties of new products. With the further enhancement in the fields of ML and DL, along with a fast increase in computational power, ML models have entered and been applied in many different fields, including materials ’performance evaluation. Through data feature extraction and the fitting of data correlations, ML models can quickly learn to acquire new skills or new knowledge to efficiently and accurately predict mechanical performance and explore new components or structures better than the training data in the design space. Not only does it reduce the wastage of resources caused by repeated trials, but it also improves the efficiency of screening and searching for new materials and structures. English is correct. Article is not very novelty, but clearly describes what it applies to and what the method is used for.
I believe that the development of each new model with the use of artificial intelligence to predict material data and its verification using a physical experiment gives a great contribution to the development of the field and innovative methods used in it.
Reviewer 2 Report
The article is clearly and concisely processed and deals with current and interesting issues.
I recommend only correcting and unifying the fonts in the figures (apart from another font, the font in some images is hard to read). Figure 4 is completely illegible.
Reviewer 3 Report
This study provides a review on machine-learning performance prediction in the field of material science. However, there are a few suggestions for the authors to consider in order to make the paper even better.
1) This paper should be extended further to cover more information related to the topic.
2) Authors are encouraged to provide an overview of machine-learning models; model structures and methodologies may be described. The performance of the models' predictions might be explored in further depth.
3) Additional recent research on the use of deep learning might be included in this study. Some papers are recommended for citation.
https://doi.org/10.1016/j.tws.2021.108076
https://doi.org/10.1016/j.istruc.2021.05.096
4) Regarding applications of machine learning, authors are suggested to divide this section into three subsections: degradation detection, material properties, and nanomaterials.
5) Can machine learning techniques be used to investigate novel materials or material design?
6) There should be a more comprehensive and analytical overview of the existing research.
Round 2
Reviewer 3 Report
Thanks for addressing my comments. The paper can be accepted for publication.